# Effect of Neuroprotective Magnesium Sulfate Treatment on Brain Transcription Response to Hypoxia Ischemia in Neonate Mice

**DOI:** 10.3390/ijms22084253

**Published:** 2021-04-20

**Authors:** Bérénice Le Dieu-Lugon, Nicolas Dupré, Céline Derambure, François Janin, Bruno J. Gonzalez, Stéphane Marret, Arnaud Arabo, Philippe Leroux

**Affiliations:** 1INSERM-UMR1245, Team 4, Epigenetics and Physiopathology of Neurodevelopmental Brain Lesions, Faculté de Médecine et de Pharmacie, Normandie Université Rouen, 22 Boulevard Gambetta, 76183 Rouen, France; berenice.le.dieu.lugon@orange.fr (B.L.D.-L.); dupre_nicolas@hotmail.fr (N.D.); francois.janin@univ-rouen.fr (F.J.); bruno.gonzales@univ-rouen.fr (B.J.G.); stephane.marret@chu-rouen.fr (S.M.); 2INSERM-UMR1245, Team 1, Genetic Predisposition to Cancer, Faculté de Médecine et de Pharmacie, Normandie Université, 22 Boulevard Gambetta, 76183 Rouen, France; celine.derambure1@univ-rouen.fr; 3Neonatal Pediatrics, Intensive Care Unit and Neuropediatrics, Rouen University Hospital, 1 Rue de Germont, 76031 Rouen, France; 4CURIB, CURIB, Faculté des Sciences et Techniques, Normandie Université, Place Emile Blondel, 76130 Mont-Saint Aignan, France; arnaud.arabo@univ-rouen.fr

**Keywords:** neonate brain, hypoxia–ischemia, magnesium, transcriptome, neuroprotection, preterm, mouse

## Abstract

MgSO_4_ is widely used in the prevention of preterm neurological disabilities but its modes of action remain poorly established. We used a co-hybridization approach using the transcriptome in 5-day old mice treated with a single dose of MgSO_4_ (600 mg/kg), and/or exposed to hypoxia-ischemia (HI). The transcription of hundreds of genes was altered in all the groups. MgSO_4_ mainly produced repressions culminating 6 h after injection. Bio-statistical analysis revealed the repression of synaptogenesis and axonal development. The putative targets of MgSO_4_ were Mnk1 and Frm1. A behavioral study of adults did not detect lasting effects of neonatal MgSO_4_ and precluded NMDA-receptor-mediated side effects. The effects of MgSO_4_ plus HI exceeded the sum of the effects of separate treatments. MgSO_4_ prior to HI reduced inflammation and the innate immune response probably as a result of cytokine inhibition (Ccl2, Ifng, interleukins). Conversely, MgSO_4_ had little effect on HI-induced transcription by RNA-polymerase II. De novo MgSO_4_-HI affected mitochondrial function through the repression of genes of oxidative phosphorylation and many NAD-dehydrogenases. It also likely reduced protein translation by the repression of many ribosomal proteins, essentially located in synapses. All these effects appeared under the putative regulatory MgSO_4_ induction of the mTORC2 *Rictor* coding gene. Lasting effects through *Sirt1* and *Frm1* could account for this epigenetic footprint.

## 1. Introduction

Magnesium appears to be a potent tool for the prevention of cerebral palsy and resulting disabilities when administered as MgSO_4_ to women in danger of preterm delivery before 33 completed weeks of gestation (GWs) [1,2,3,4]. Historically, MgSO_4_ used as a tocolytic did not prove to have efficiency in retarding delivery, but retrospectively it has shown neuroprotection effects at low doses [5,6]. The Mg^2+^ ion has innumerable biological targets e.g., as co-factor in nucleic acid biology, protein synthesis, enzymatic systems, or protein–protein interactions [7]. Its putative effects resulting in fetal neuroprotection are therefore very numerous, and its actual modes of action remain elusive. Given at neuroprotective (low) doses, MgSO_4_ has no deleterious side effects, which is a requirement for prophylactic treatment. However, owing to its large spectrum of putative effects, it appears necessary to strengthen the evidences of its harmlessness at neuroprotection doses, since high doses of MgSO_4_ have adverse effects at the cellular and whole-organism levels [8,9].

Pediatricians have detected diverse effects of prenatal MgSO_4_ in newborn presentation and behavior. Reports on cardiorespiratory function were somewhat controversial, showing no effect or depressant effect of antenatal magnesium [10,11]. It reduces heart rate but stabilizes blood pressure and reduces cerebral blood flow [10,12]. Overall, prenatal MgSO_4_ was shown to decrease the need to resort to intubation in very preterm births [13]. Finally, although MgSO_4_ supply has not been assessed as protective against bronchopulmonary dysplasia, hypomagnesemia was described as a risk factor [14].

Magnesium is mainly restrained in intracellular stores and circulating levels after bolus injection rapidly decrease due to near saturation of renal reabsorption pumps in physiological magnesemia conditions [15]. Antenatal magnesemia is high in the preterm and bolus MgSO_4_ resulted in a peak observed at delivery followed by a moderate hyper-magnesemia during the first 3 days of life in very preterm infants exposed in utero [16,17]. In neonatal mice, the clearance of Mg^2+^ appeared even faster since no hyper-magnesemia remained 12 h after a 600-mg/kg bolus injection [18]. Thus, the presence of MgSO_4_ in the mother and fetus organisms after a low bolus administration is fleeting at the blood level, whereas lasting neuroprotection is expected and in fact observed in the subsequent hours and days [15,19]. It is possible that intracellular stores are rapidly sequestrated, although this requires active and saturable mechanisms. One may therefore consider the possibility that transient Mg^2+^ elevation initiates some processes that predispose the fetus against prospective noxious experiences, such as preterm birth and related consequences, somewhat as may proceed a pre-conditioning. Indeed, in animal studies, Mg^2+^ supplementation prevents maternal inflammation and offspring cerebral injuries [20,21,22].

It is likely that prenatal Mg^2+^ supplementation exerts neuroprotection via systemic and central effects. It has been suggested that direct effects at the fetal brain level occur through calcium antagonism and/or glutamate N-methyl-*D*-aspartate type glutamate receptor (NMDA) blockade [23]. Inflammation in the mother or consecutive hypoxia–ischemia is a major trigger of neonatal brain lesions and magnesium exerts anti-inflammatory activity in human or animal neonates [24,25]. MgSO_4_ pretreatment provided neuroprotection in the Rice and Vannucci experimental paradigm modeling hypoxia-ischemia (HI), and in the excitotoxic model of ibotenic acid injection [26,27]. In these models, the effects of MgSO_4_ do not account for maternal-mediated effects, suggesting direct effects in the neonate brain and improved delayed behavior [18,28]. In the 5 day-old mouse brain, HI provoked an early onset of coordinated gene inductions (repressions) 3 h to 12 h after insult, affecting transcription regulation at the RNA-polymerase-II complex, inflammation, cell death, and angiogenesis. In parallel, convergent gene repression affected many genes encoding for cholesterol metabolism enzymes and synaptic actors [29]. The causal factors of long-term deleterious consequences among these regulations are likely to be explored in the field of inflammation as well as its interaction with development [30,31,32].

Magnesium may affect a plethora of factors and interferes with HI [7,33,34]. In light of our previous description of the strong effects of HI on the transcription pattern in P5 mice, the Rice and Vannucci model appeared to be a simple approach for an overview of mechanisms recruited in sole treatment with MgSO_4_, and of its modulation of HI-effects [29]. Neonatal mice at 5 days post-birth (P5) are usually considered a suitable model to explore brain development and injuries of human preterm born between 28 and 34 GW [35,36]. A further demonstration that the insult sensitivity of P5 and P10 mouse brains accurately mimicked preterm neonates and term infant brains was obtained through magnetic resonance imaging and histology after HI using the Rice and Vannucci model. In fact, P5 mice only showed white matter hypersignal and microglial activation. Moreover, age-dependent memory or social behavior defects were observed in adults [28,37].

The period of special white matter vulnerability is the recommended therapeutic window of MgSO_4_ use (before 33 GW) for preterm prophylactic neuroprotection [38,39]. This study had dual goals—(i) the description of the proper effects of MgSO_4_ single bolus injection at P5 in mice, investigated at the transcription level in short delays and at the behavior level in grown-up mice, (ii) the identification of the MgSO_4_ interference with HI-induced transcription effects and with spontaneous developmental transcription evolution.

## 2. Results

Transcriptome analyses were performed on the basis of three independent experiments. Experiment 1 compared the effects of single MgSO_4_ injection to phosphate buffered saline (PBS) injection, Experiment 2 compared the effects of MgSO_4_ or PBS injection prior to HI, and Experiment 3 compared the effects of HI compared to control mice (Figure 1). The comparison of the 3 experiments was performed using the combination of two-color and one-color analyses of hybridization data detailed in Section 4.3 (Figure 2).

### 2.1. Effects of Single-Injection MgSO_4_ on Transcription

A total of 1411 genes exhibited significant alteration of expression, at least at one time point from 90 min to 24 h after injection. Two-fold more repressions (*n* = 973) than inductions (*n* = 447) were recorded. Nine genes appeared in the induction and repression lists; with seven at the 6 h time point, revealed by separate probes, suggesting alternative splicing (*Atp1a3, Aut2, Celf4, Kalrn, Psd3, Tenm4, Tln2*), and two exhibiting successive opposite variation (*Dmtn, Septin3*) (Appendix A). The time course study revealed a wave of transcription inductions and repressions peaking after 6 h and including 1322 genes (93.7% of the total list in 24 h) (Figure 3A). The median fold change (FC) amplitude was significantly lower for inductions (1.68; Q1 = 1.64−Q3 = 1.80) than for repression (−1.82; Q1 = −2.02−Q3 = −1.65; *p* < 0.0001; according to the Mann–Whitney test) (Appendix A).

#### 2.1.1. Up Keywords

The highest enrichments of UP_keywords (seKws) specific to list A were noted for “synapse,” “neurogenesis,” and “cell junction” (Table 1). Looking at neurogenesis one could note many genes coding transcription factors, membrane associated scaffolding and cue proteins, neurotrophin transmission associated proteins. The cell junction list included all seKws associated to synapses and had 20% of genes in common with the cytoskeleton list. Enrichment of these seKws, also exhibiting very low false discovery rate (FDR) values, clearly indicated that MgSO_4_ targeted the transcription in neurons, inducing the repression of membrane proteins and especially of synaptic proteins. Many glutamatergic synapse elements exhibited high amplitude repression (*Grin1, Shank3, Dlgap3, Dlgap1,* and *Slc17a7* coding the Vglut1 transporter). In addition, *Grina3, Dlg2, Gria1, and Grik5*, also appeared to be repressed, but with less amplitude, whereas *Grik3* was induced. Repression of *Ache* and the muscarinic receptor M1 genes were also recorded (see details in Appendix A).

#### 2.1.2. GO Analysis

DAVID^®^ Gene Ontology identified 44 seGO-terms; 10 biological processes (BP), 24 cell components (CC), and 10 molecular functions (MF) subclasses. Five seGOterms were related to neurons, synapses nervous system development, and postsynaptic density, exhibited the highest significant *p*-values and showed among the highest number of genes involved (Table 1). With a lower significance seGO-terms also focused on cell junctions, neuron projection or ion binding, and with decreasing incidence on, kinase-phosphorylation activities, the cytoskeleton, the regulation of transcription, cell signaling, and the Golgi apparatus (Figure 3B, Appendix A). Those genes associated with seGO-terms mainly underwent repressions.

#### 2.1.3. Pathway Analyses

##### KEGG Pathways

DAVID^®^ analysis allowed the extraction of 45 significantly enriched Kyoto Encyclopedia of Genes and Genomes (KEGG) pathways (seKpaths) at the Expression Analysis Systematic Explorer (EASE) *p*-value threshold < 5 × 10^−2^ (and 14 others out of scope) (Appendix A). Of note the lowest *p*-value recorded were associated to axon guidance (mmu04360, *p* = 1.66 × 10^−5^ in Bonferroni test), thyroid hormone signaling, neurotrophin signaling, cAMP signaling, and pathways in cancer (including genes involved in extracellular matrix–cell regulation and insulin signaling (Table 1). Less significant seKPaths were related to neurotransmission systems (glutamatergic synapse, GABAergic synapse, retrograde endocannabinoid signaling, cholinergic synapse, and many peptide signaling and membrane transduction pathways (Figure 3C, Appendix A).

##### Ingenuity Pathway Analysis (IPA) Pathways

Pathway extraction using IPA was based on a strictly different approach and allowed a much broader view as the strategy did not depend on previous validations, as for the KEGG pathways, which are relatively limited in mice. IPA pathways were extracted at all five time points on the corrected log ratio of the 1492 probes affected by MgSO_4_ injection at least at one time point. A total of 13 significantly enriched IPA pathways (seIPaths) exhibited Fisher’s Exact Test *p*-value < 1 × 10^−3^, of which 11 had z-score < −2 (Table 1, Figure 3C).

Se-IPath enrichments were noted early (1 h 30 min) and exceeded what could be expected from the gene kinetics record (Figure 3D). The seIPaths exhibiting the most significant *p*-values, together with more specific genes included were synaptogenesis, reelin signaling in neurons, and axonal guidance (Table 1). Only one seIPath had a positive *p*-value (PTEN signaling) indicating an activation of the pathway, although this was exclusively based on gene repressions (Appendix A).

##### IPA^®^ Upstream Regulators

Putative upstream regulators (pURs) refers to key regulators, on the basis of convergences between reports in the literature of any relationships between this gene/protein expression/activity and the enrichment of these anticipated effects in an experimental set of data. A z-score, significant when >2 (or <−2), indicates the sense of the regulation. When a pUR was not affected at its transcription level, one must conclude that the experimental condition had downstream effects on its translation or on the function of the coded protein. In the following text, pURs are identified using protein symbols, whereas gene symbols (in italics) refer to transcription observations. Furthermore, the terms induction and repression are devoted to transcriptions observation, whereas the effects of pUR are called activation and inhibition.

Only two pURs (Mnk1 and Frm-1) showed very highly significant *p* value (<1 × 10^−7^), although MgSO_4_ did not affect their transcription. Upstream analysis revealed inhibition of Mnk1 and activation of Fmr-1, but the putative effects of MgSO_4_ on downstream effects converged toward essentially the same canonical pathways, i.e., the inhibition of synaptogenesis, reelin-, and glutamate receptor signaling (Figure 4, Appendix A). The mechanism of the interaction of Mg^2+^ with Mnk1 and Fmr-1 remains elusive. Among the 30 genes putatively downstream to Mnk1 in the Mg group, three genes affected in HI showed reversed effects in MgHI (*Map1b, Mapre2, Vim*), 26 genes affected by Mg no longer appeared in MgHI, and the four genes remaining exhibited inverted regulation (*Bag3, Cplx2, Trim32, and Ttc3*).

As a polyribosome-associated mRNA-binding protein with specific activity in brain development and synapse plasticity, Fmr-1 is an indicator of Mg tropism toward protein translation in neurons. Sirt1 was another, although less significant pUR, with lasting occurrence, a positive z-score, and *p* < 1 × 10^−4^ at 24 h, which is involved in histone de-acetylation (Appendix A).

##### Synthesis

MgSO_4_ injection mainly provoked gene repressions, which predominated even more significantly in the coordinated effects revealed by seKPaths, seIPaths, and pUR, indicating a convergent expression of activity toward two groups of functions (Figure 3B–D, Appendix A).

Both seKPaths and seIPaths were related to nervous system development (axon guidance, thyroid hormone signaling, synaptogenesis signaling, reelin signaling, netrin signaling). pUR analysis pointed out Mknk1 and Fmr1 as MgSO_4_ targets for the cooperative inhibition of synaptogenesis, reelin, and glutamate signaling. Repressions largely predominate in these pathways, suggesting a transient brake in nervous system development (Table 1). pUR analysis suggested Frm1 as an MgSO_4_ target involved in the inhibition of synaptogenesis as well as implicating Sirtuin-1 as a potential site for Mg epigenetic activities.

A large group of seIPaths showed effects of MgSO_4_ effects on chemical signaling by hormones (insulin, corticoliberin, renin-angiotensin, GnRH) and neurotransmission modulators (NO, Ephrin, opioids, endocannabinoids). Transduction systems were also affected (signaling by Rho GTPase and CREB). All were slowed down (z-scores < −2) at several time points indicating reduced activity.

MgSO_4_ affected the expression of 299 developmentally regulated genes, mainly anticipating ontogenetic increases, and to a lesser extent ontogenetic decreases of expression (Appendix A). Very few MgSO_4_ effects were antagonistic to ontogeny (Appendix A).

### 2.2. Long-Term Behavioral Effects of Neonatal MgSO_4_ Single Injection

Regarding the effects of MgSO_4_ single injection on the expression of synaptic genes and spontaneous ontogenic evolution, there may be a concern that it could affect the brain development and result in lasting behavioral alterations. MgSO_4_ effects were examined in sex-separated groups in several tests, in comparison with positive control animals treated at P5 with MK-801 or ketamine. Detailed effects and statistics are given in Appendix A.

#### 2.2.1. Spontaneous Locomotor Activity

The activity estimated by the running distance in the free wheel showed a significant effect of sex (females were more active than males) in the two-way ANOVA. The Bonferroni post-test showed an increased activity in females treated by ketamine (Figure 5A). No differences between acutely treated neonates (BPS, MK-801, or MgSO_4_) nor in Repeat-edly treated animals (PBS or MgSO_4_) were observed. It is noteworthy that repeated injection, which required daily handling in pups, had a lasting reduction effect on activity, although the nature of the treatment had no effect (Appendix A).

#### 2.2.2. Open Field

##### Animal Spontaneous Activity

Open field exploration was recorded for 30 min in the open field and was analyzed in acute or repeated treatment groups using two-way ANOVA (sex/treatment). Sex differences were observed but no effects of treatments on the distance run through the open field or on the exploration time of the central area (Appendix A).

##### MK-801 Evoked Hyperlocomotion

MK-801 acute injection at the end of 30 min exploration induced locomotor activation in adults in all neonatally treated groups. It induced running activity and stereotypies but reduced vertical exploration (Appendix A). The post-test revealed that neonatal MK-801 increased vertical exploration in males (*p* = 0.0116) and had a tendency to produce the same effect in females (*p* = 0.0921) (Figure 5B). An interaction of sex with treatment was observed in stereotypies. The post-test revealed an amplification of stereotypies induced by acute MK-801 in males (*p* = 0.0018) whereas it had a tendency to reduce them in females (Figure 5C).

#### 2.2.3. Social Interaction

MgSO_4_ had no influence on adult behavior nor did MK-801 (Figure 5D,E). Global activity evaluated on the basis of total distance run or rearing, did not reveal differences related to acute nor repeated treatments although sex differences were observed (not shown). The time spent in the contact zone was selected as a raw index of social interaction and did not show an effect of treatment (Appendix A). Tenuous treatment effects were detected using the comparison of inter-zone crossings to evaluate the activity induced by the apparition of the congener. Treatments had no effect on raw indexes according to two-way ANOVA (on sex and treatment), in acute or repeated treatment groups whereas sex effects were constant (Appendix A). The distribution of crossings did not differ between groups during habituation whereas it revealed a significant difference in females during the interaction period (Figure 5D,E, Appendix A). Neonatal ketamine affected activity provoked by an unknown congener in females (Chi^2^ = 31.15_df3_, *p* = 0.0003). The reduced number of entries evoked by the congener did not necessarily reflect a poor interest for the congener since time spent in the contact zone was not reduced in these ketamine-treated females (Appendix A)). Thus, the experimental paradigm appeared to be discriminant, and did not show any effect of MgSO_4_ on this behavior, even after repeated neonatal administrations.

#### 2.2.4. Synthesis

MK-801 or ketamine, used as positive controls, induced sex-dependent effects in the different behavior tests. This selectivity of MK-801 and ketamine among tests and sex reinforces the choice of the tests as they appeared to be relevant for distinct behavioral traits. In addition, they showed ketamine effects that were independent of glutamate NMDA receptor subtype blocking, since they were distinct from the effects of MK-801. The question of long-term effects of single neonatal ketamine exposure is once again addressed. Of note, neonatal acute or even repeated MgSO_4_ neonatal single injection did not alter any of this series of independent behavioral traits (Appendix A).

### 2.3. Interaction of MgSO_4_ with HI-Induced Transcriptional Modulation

#### 2.3.1. General Observations

The modulation by MgSO_4_ of the effects of HI did not result from the differences observed in experiment 2; therefore, we must take into account the effects of HI unaffected by MgSO_4_ and reciprocally the MgSO_4_ effects observed after HI (Figure 2).

The MgHI effects did not appear as a superposition of the separate effects of MgSO_4_ and of HI. The major proportion of regulated genes was specific to one of the three experimental conditions (two-color analyses of experiments 1–3). The list of MgHI vs. HI effects (list ②) showed only 145 genes in common with that of Mg effects (List ①), and 89 genes in common with that of HI effects (List ③) (20.1% and 12.8% of ②, respectively). A similar observation was made when comparing MgHI vs. Ctrl (List ⑦) in one-color analysis (Figure 6A).

MgSO_4_ pretreatment induced more negative regulations than enhancement of expressions due to HI in the P5 mouse brain (513 vs. 180 in list ②). It modified the transcription response as early as 3 h after HI, i.e., before showing proper transcription effects (peak at 6 h), indicating that before magnesium had transcriptional effects, it affected effector systems (e.g., enzyme activities), which in turn largely modified the whole transcription response to HI. Much fewer genes continued to have an altered expression 12 h after MgHI, and most showed a reduction of an effect already seen at 3 h, indicating the short-term biological response (Figure 6B).

MgSO_4_ also evoked de novo HI-inductions (*n* = 545) and repressions (*n* = 943) revealed by one-color analyses (List B_1_), confirming that MgSO_4_ rather than preventing the HI response, oriented the HI transcription response in a new direction (Figure 6C).

The assembly of MgSO_4_ effects in HI (List B) results from the summation of de novo inductions (repressions) compared to HI (List B_1_), as well as reversions (List B_2_), and amplifications of HI effects (List B_3_). From these the effects of MgSO_4_ that were unchanged in the MgHI groups (list A_1_) must be subtracted. Thus, the analysis was complex, as the different lists resulted from different kinds of data extraction. List B describes MgSO_4_ modulation of the HI transcription response. A total of 1964 genes in MgHI (842 inductions, 1091 repressions and 31 biphasic evolution or putatively spliced) showed effects that differed from HI effects. Repressions predominated (55.6%), much more than in HI effects in list ③ (30.7%), but somewhat less than in List A of MgSO_4_ proper effects (68.5%) (Appendix A). The comparison of List B with HI effects (List ③) revealed a minority of genes regulated in both conditions (171 on 723 in ③), and 1488 genes affected de novo (Appendix A).

A small proportion of HI effects (18.2%) were unaffected by MgSO_4_ pretreatment (List C) (Figure 6C). Even based on the restrained gene series presented in list C, David^®^ analysis showed the persistence of major HI effects on BP seGOterms (response to cytokine, transcription from RNA polymerase II promoter, cytokine chemotaxis, DNA-templated transcription, positive regulation of gene expression, and negative regulation of neuron apoptotic process) (see below).

#### 2.3.2. DAVID^®^ Biostatiscal Analyses

##### Up Keywords

David^®^ extraction identified 18 seKws from List B, including 1576 genes (80.2% of the total List B). The most enriched seKws were mitochondrion (*n* = 176), mitochondrion inner membrane (*n* = 45), and electron transport (*n* = 22), ribonucleoproteins (*n* = 54) and ribosomal proteins (*n* = 38) (Table 2). Repressions predominated in mitochondrion- and ribosome-seKws (77.2% and 90.7%, respectively). Conversely, inductions in seKws were mainly related to ubiquitination, transcription, apoptosis, and transport (Appendix A).

Four seKws were enriched from both List A and List B, referring to basic mechanisms (phosphoproteins, alternative splicing, Ubl-conjugation and transcription regulation). These few seKws included hundreds of genes in A or B, but about 10% of these genes were common to Lists A and B, 7–8% of which even behaved in opposite direction. Many of the seKws from List B were also enriched, due to de novo effects (List B_1_) and consisted of repressions (Appendix A).

Reversion by MgSO4 of HI effects (Lis B_2_) exhibited seKws in the fields of the major effects of HI; i.e., inflammation, immunity, and regulation of transcription by RNA-polymerase II. Whether these effects underlie neuroprotection by MgSO_4_ requires further studies.

Altogether, these observations indicate that MgSO_4_ did not so much change the nature of the HI transcription response, but rather it profoundly modified tissue adaptation to injury.

##### GO Analysis

DAVID^®^ extraction of seGO-terms from List B identified eight CC-seGO-terms and two MF-seGO-terms (Table 2). The terms “mitochondrion” and “ribosome” exhibited high counts and the lowest *p*-values (<1 × 10^−20^), although their enrichment were modest (Appendix A). Most of these terms appeared from list B1 of de novo effects, indicating that MgSO_4_ pretreatment allowed HI to affect mitochondria and ribosome related genes. The question of whether effects at these sites could account for neuroprotection by MgSO_4_ would be worth investigating. Nevertheless, and even if they were scarce, enriched BP and MF seGO-terms do not indicate major modification in cell functions.

##### KEGG Pathways

Two seKpaths were extracted from List B (*p* < 1 × 10^−4^ in Fisher exact test and FE > 2); ribosome and Parkinson’s disease, each including 30 genes (Appendix A).

#### 2.3.3. IPA Biostatistical Analyses

##### IPA Pathways in MgHI Mice

IPA^®^ analysis of List B could not be performed as this list resulted from lists obtained using two-color and one-color data, in distinct Genespring^®^ standardizations. However, separated approaches toward the MgSO_4_ interference with HI transcription were carried out in two-color MgHI vs. HI (List ②) and one-color MgHI vs. Ctrl (List ⑦) analyses (Figure 7).

Only one seIPath (reelin signaling in neurons) reached the 1 × 10^−4^
*p*-value from List ② at 3 h (Appendix A). Several pathways (i.e., synaptogenesis signaling pathways) had *p*-values below 1 × 10^−2^ and significant z-scores (<−2), although they included modest gene numbers. Of note reelin and synaptogenesis signaling pathways were also enriched by the effects of MgSO_4_ alone. No seIPaths were extracted from list ② at the 12-h time point.

IPA^®^ analysis of MgSO_4_ effects on HI response evaluated based on one color analysis (List ⑦) revealed only one seIPath with *p* < 1 × 10^−4^ (the sirtuin signaling pathway), although with no clear indication of regulation sense (z = 1). Less significant, although clearly inhibited, was EIF2 signaling which included 26 genes (*p* = 1.82 × 10^−4^, z = −3.051). Repression (more than 75%) affected 15 ribosomal proteins, indicating an inhibition of protein translation in MgHI conditions. This inhibition of translation appeared de novo after HI and MgSO_4_ pretreatment since 28 of these genes plus four others, associated to EIF4 signaling, were not affected in the absence of pretreatment (HI in list ③). Other de novo pathways were detected, i.e., the lasting repression of glutathione redox reactions I, and melatonin- and CDK5-signaling (Appendix A).

##### IPA^®^ Upstream Regulators in MgHI Conditions

The identification of pUR in the experimental condition combining MgSO_4_ treatment and HI would not reveal a target, as would be the case in single pharmacological conditions. Rather, it would indicate putative hubs affected by the combined effects of the two stimuli which would not represent a simple summation of effects. No pUR could be extracted from the two-color analyses of MgHI vs. HI effects at 3 h or at 12 h, possibly because of small sizes of the lists. pUR extraction from one-color analysis of MgHI vs. Ctrl (List ⑦) identified three pURs at highly significant *p* values < 1 × 10^−4^; two pURs were activated (Rictor and Creb-1) and one was inhibited (Tfrc) (Figure 8A, Appendix A).

### 2.4. Tracking of Magnesium Targets

We compared, seIPaths in nine separate conditions—MgSO_4_ effects at 1 h 30 min, 3 h, 6 h and 12 h, HI effects at 3 h and 12 h; MgHI vs. HI effects at 3 h and 12 h (List ②) and MgHI vs. Ctrl effects using the one-color approach (list ⑦) (Figure 7).

#### 2.4.1. Effects of MgSO_4_ Injection

Fmr1 and Mnk1 were two pURs among the effects of MgSO_4_ (Figure 4 and Appendix A). In Mg, 23 of the 30 genes under the Mnk1putative upstream inhibitor were specifically associated to this pUR. They were not significant pUR in HI, but Fmr1 and Mnk1 showed inverted z-score and nearly significant *p*-values in the one-color MgHI analysis. Reciprocal activation of Fmr-1 as a pUR in Mg appeared far less significant in MgHI at 3 h and was possibly reversed in one-color analysis (Appendix A). Owing to its recurrent and highly significant enrichment, it appears to be a highly likely target on which MgSO_4_ had an inhibitory effect, downstream to transcription. However, as almost no pUR downstream gene were affected in HI, the regulation on Mnk1 and Fmr1 by MgSO_4_ has probably no direct effect on neuroprotection.

Casr, Sirt1, and IL4 appeared to be lowly significant pURs in mice treated solely with MgSO_4_. All three were highly significant potential hubs in HI but these putative activities did not remain in MgSO_4_-pretreated animals (Appendix A).

#### 2.4.2. pURs Extracted from the Panel of Genes Affected after HI

IPA^®^ extraction allowed us to identify 48 pURs with *p*-values below 1 × 10^−4^ (Appendix A). A high number of pURs in HI had disappeared in MgHI group. As the majority of pURs in HI were related to inflammation, acute reaction and transcription one could expect MgSO_4_ effects to point in these directions, although we could not identify specific targets at this point. Only three of these 48 pURs appeared in the Mg group (Crem, Casr and Sirt1), and did not remain in MgHI. However, it seems unlikely that these effectors are MgSO_4_ targets since the majority of their downstream-targeted genes had expression patterns that were unchanged in MgHI, compared to HI conditions (31/54) (Appendix A).

#### 2.4.3. Effects of Combined MgHI Conditions

Many seIPaths appeared to be affected with different kinetics, in opposite sense in Mg or HI groups (Appendix A). Among them, several showed Mg-like or HI-like effects in MgHI, whereas others disappeared. The inversion of HI activation in MgHI was noted for oxidative phosphorylation, ephrin receptor signaling, and B-cell receptor signaling (Figure 7). Abolitions of HI activation were noted for mTOR signaling, hepatic fibrosis signaling, and ILK signaling. The highly significant inhibition of the super-pathway of cholesterol biosynthesis observed 12 h after HI also disappeared in MgHI.

Several apparent discordances appeared between MgHI vs. HI effects and MgHI vs. Ctrl effects, likely due to the different timing of evaluation, and the occurrence of transient responses (Figure 7). Indeed, the rapid bio-availability of Mg^2+^ and its rapid clearance may have short-term effects that do not last or are overwhelmed by HI effects. Thus, comparing pURs in the nine lists, we could classify the observations in four groups. Only one gene/protein (Rictor) appeared as a de novo pUR in MgHI. Three pURs exhibited opposite z-scores in the Mg and HI groups, resulting in annihilation or HI reversion effects in MgHI (Casr, Ppara and Fmr1). In addition, larger series with less clear patterns regrouped pUR in HI for which MgSO_4_ prevented (*n* = 68) or did not affect (*n* = 48) their significance in MgHI (Appendix A). These series contain many genes involved in effectors in inflammation, cell death, transcription, and DNA repair, indicating that MgSO_4_ in fact reduced these effects, although the analysis pointed out few specific putative Mg targets for these effects. Five entities, however, retained our attention—Rictor, Infg, Il4, Ppara, and Sirt1.

*Rictor* expression was unaffected in all conditions. mTORC2, coded by *Rictor* appeared to be a de novo pUR, based on de novo gene regulations in MgHI conditions. It had the lowest *p*-value (4.70 × 10^−6^), a high z-score (4.885), and the highest number of downstream targeted genes (*n* = 37 genes) (Figure 8A). The majority of these genes (32) participated in few canonical pathways—mitochondrial dysfunction; oxidative phosphorylation, protein ubiquitination, mTOR signaling and regulation of elongation by Eif2 and Eif4. These functions clearly confirmed the aforementioned seKws, seGO-terms, KEGG, and IPA pathways observations, indicating that in MgHI condition, protein translation and oxidative metabolism were inhibited through the repression of ribosomal proteins, ubiquinone oxydoreductases, cyclooxygenases, and proteasome proteins (Figure 8A).

Sirtuin1, appeared very early (1 h 30 min) and showed lasting activation of pURs in MgSO_4_ treatment based on the regulation of 32 putative downstream genes, mainly repressions. An opposite activating pattern was observed 12 h after HI. Sirtuin1 did not appear to be a pUR in the MgHI one-color list, but the Sirtuin signaling pathway was a significant pathway extracted from the de novo list ⑦. These observations indicate that MgSO_4_ influenced the expression of many genes downstream to Sirtuin1, possibly at the protein level, resulting in a change in HI induction in pretreated animals (Figure 8B).

Infg was the most significant pUR in HI conditions at 3 and 12 h as a potential activation factor (z > 4.4). Although a high z-score was measured in the MgHI one-color analysis, it did not remain a significant pUR (Figure 9A). A majority (*n* = 30) of the 42 genes downstream of Ifng in HI at 12 h were no longer affected in MgHI conditions. Many other factors involved in inflammation processes showed similar patterns (Appendix A). pUR comparison did not allow to extract a unique target for MgSO_4_ in lowering the HI-induced inflammation process, but the global tendency allowed us to conclude that MgSO_4_ prevented inflammatory effects of HI in the P5 mouse brain as a possible cause of its neuroprotection effects.

Il4 appeared to be an inducer in HI (z = 2.0333 h after HI), and a possible inhibitory pUR of MgSO_4_ effects (z < −2 from 3 to 12 h), but did not reach significance in MgHI conditions (11 of 16 genes downstream to Il4 affected in HI were unaffected in MgHI) (Figure 9B). The *Il4* gene did not show transcription regulation in any experimental.

Ppara appeared to be a lowly significant activating pUR in HI at 12 h (*p* = 4.02 × 10^−4^, z = 2.302), and did not appeared significant in MgHI at 3 h (*p* > 5 × 10^−2^, z = −1.938) (Figure 9C). It retained our attention however, as it may be a hub for the inhibition of the cholesterol biosynthesis pathway in HI at 12 h. (Figure 9C). Two of these 5 genes (*Idi1* and *Msmo1*) were activated in MgHI, while *Mvd, Mvk,* and *Nsdhl* were unaffected. These divergent effects preclude MgSO_4_ effect on Ppara.

#### 2.4.4. HI Effects Insensitive to MgSO_4_

seGO-term extraction from List C enabled us to evaluate the putative effects of HI that were insensitive to MgSO_4_ pretreatment. Gene inductions, predominant in HI, are related to transcription by RNA-polymerase-II, inflammation and angiogenesis as previously described [29]. Reciprocally, the repression of genes encoding cholesterol biosynthesis enzymes was observed (*Idi1, Mvk and Mvd, Msmo1 and Nsdhl*), although the significance threshold was not reached. A minority of genes affected by HI in list ③ appeared to be unaffected in List C—166/499 inductions (33%) and 66/224 repressions (29%). As far as gene ontology is valid for partial lists, seGO-terms related to inflammation were absent in C, with the exception of the response to cytokine and monocyte chemotaxis. Otherwise the seGO-terms related to RNA polymerase II identified from List ③ remained in List C (Appendix A).

## 3. Discussion

Neonate mice at P5 show forebrain development and sensitivity to insults which was somewhat representative of human brain development and vulnerability around 28–32 GWs, and may be used as experimental surrogates [24,35,36,37]. However, one must keep in mind that putative MgSO_4_ neuroprotection in humans also results from activities in the mother, at the placental level, and that the purpose of the present study was restricted to MgSO_4_ effects on the neonate. Acute single MgSO_4_ 600-mg/kg administration in P5 mice evoked transient transcription effects, mainly repressions, for a short period culminating 6 h after injection. The gene ontology and pathways extraction pointed out the effects of MgSO_4_ on synapses and brain development (Figure 10). The administration of 600-mg/kg MgSO_4_ in P5 mice induces a transient peak concentration 30 min after administration. Blood concentration was 1.5 mM after 6 h, a concentration detected for longer periods in preterm born from mother having received 4–6 g bolus plus 1–2 g/h maintenance [18,40]. In mice pups, the concentration returned to basal level within 24 h, while it often remained above 1 mM in human, leading us to consider the 600-mg/kg bolus as a low MgSO_4_ dose [17,40]. Under these conditions MgSO_4_ did not induce apoptosis [28].

Concerning the transcription effects of MgSO_4_ alone, one could note the convergence toward synapse differentiation and function in either, seKws, seGO-terms and pathways identified in Mg groups. Two highly significant and durable pURs were identified as potential targets of Mg^2+^: Mnk1 and Frm-1. These two proteins have activity in protein translation. Mnk1 requires Mg^2+^ binding on a specific binding loop to achieve maximum activity [41]. Mnk1 inhibition as a pUR nevertheless appeared to be the most significant and durable putative target of MgSO_4_, inducing the actual repression of many downstream genes. *Fmr1* transcription was not affected in any experimental group but the synaptic functional regulator Fmr-1 also appeared to be an activating pUR site for MgSO_4_, although the literature does not report an Mg^2+^ cation link to protein activity. Functional relationships between (i) MgSO_4_ effects through Mnk1 and Fmr-1, and (ii) subsequent synaptogenesis appear to be reliably supported by our finding, but the modes of action remained undetermined. As a polyribosome-associated mRNA-binding protein, Fmr-1 indicates a Mg^2+^ putative action on protein translation that appeared more clearly in the MgHI group.

Although transcription effects were very transient, it is possible that this interference with such a crucial development process could have long-term consequences, as reported for several pharmacological agents eventually used in neonatal pediatrics (corticoids anesthetics, NSAIDs), or in animal models (MK-801, ketamine) [42,43,44,45,46]. We assessed several behavioral responses in adults exposed to neuroprotective dose of MgSO_4_ compared with drugs with known effects at P5. The battery of tests was chosen to mitigate the simple spontaneous response to a new environment and more integrated drug- or congener-evoked responses. This battery of tests in fact allowed for the discrimination of non-convergent behavioral parameters, since positive controls injected with MK-801 or ketamine had separate sex- and test-specific effects. MgSO_4_ acute neonatal treatment did not modify adult behavior in these tests. Although it is difficult, from a statistical point of view, to demonstrate an absence of effect, these data argue against a putative deleterious effect of MgSO_4_ at a neuroprotective dose, in particular through its interaction with the NMDA-type glutamate receptor. Previous studies have reported that high doses of MgSO_4_ may impair vascular development, then nervous parenchyma [8]. The use of repeated MgSO_4_ administration was performed to assess this possibility, in conditions designed to mimic the prolonged administration that eventually occurs in pregnant women at risk of preterm delivery. In this respect, it is interesting to note the absence of lasting effects of repeated MgSO_4_.

The present data on MgSO_4_ effects showed transient transcription modulation apparently devoid of delayed deleterious effects and putatively capable of exerting brain protection. The study of MgSO_4_ in neonate mice confirmed its harmlessness, which has also been largely reported in humans. This does not preclude other effects of MgSO_4_, i.e., on the highly vulnerable white matter around 30 GWs (MagNUM) study; [47]. In the MagNUM study, white matter analysis by means of magnetic resonance imaging revealed the induction by MgSO_4_ of microstructure development, a finding coherent with the present observation of MgSO_4_ transcription inductions/repressions orthologous to spontaneous development in 264 genes.

MgSO_4_ is not only safe, it is also a potent prevention agent of motor, behavior, and cognitive deficiencies in preterm subjects [6,48], although its mode of action is not fully understood.

Another concern about the use of MgSO_4_ as prophylactic a neuroprotection in fragile populations was the frequent statement that it acts as an NMDA channel blocker [49]. This belief was often based on a very dated description of NMDA function and also to the fact that MgSO_4_ was neuroprotective in NMDA mediated neurotoxicity [18,35,50]. This possibility is frightening owing to the very deleterious effects in the developing brain of NMDA blockers such as MK-801 in animals [43], and with the relationship of NMDA hypo-activity or genetic defects with schizophrenia [51]. Administration of Mg^2+^ reduced the release of Mg^2+^ from NMDA channel, but it does not act as an antagonist. Mg block lifting is dependent on depolarized membrane potential but not on glutamate and glycine (D-serine) binding [52]. Under depolarization Mg^2+^ loses intra-channel binding high affinity, making it unlikely that overload would reduce NMDA excessive activation. Moreover, the expression of NMDA-R subunits in 30-GW human brains have shown Mg^2+^ blocking sensitivity [53]. In vitro studies showed that in different neuron populations, low doses of glutamate-induced pre-conditioning toward high glutamate excitotoxicity can be mimicked using Mg^2+^ supplementation, indicating a clearly more complex effect of magnesium on NMDA receptors than channel blocking [52]. At the transcription level, MgSO_4_ repressed *Grin1* and *Grin3a* in Ctrl and reversed *Grin2b* induction after HI (Appendix A). These effects together with the reduction by Mg^2+^ of glutamate release would underlie its anti-glutamatergic toxicity more likely than NMDA blockade [54]. The present behavioral observations showed that MgSO_4_ effects did not mimic the effect of the NMDA specific blocker MK-801. In addition, transcriptome IPA^®^ analysis never proposed genes coding NMDA receptor subunits as pURs, even at low significance levels before filtering. The statistical proof of an absence of effect is always a difficult challenge. Thus, a demonstration that MgSO_4_ does not affect NMDA transmission in the 30-GW fetus is probably elusive; however MgSO_4_ may be considered a safe prevention tool in the neonatal context, and is very likely devoid of NMDA-interfering side effects. On the contrary, the incidental observation of ketamine’s interference with basal activity and social behavior renews the question of its use (or abuse) in the newborn [55]. The clearly distinct effects of ketamine compared to MK-801 ruled out ketamine’s interaction with NMDA in these effects and lead us to consider ketamine for broader potential neurotoxic effects.

A major concern in very preterm care is the high incidence of brain hemorrhages. Meta-analyses of prophylactic antenatal MgSO_4_ exposure did not conclude that it led to intraventricular hemorrhage (IVH) prevention, although tendencies were reported [6]. Relatedly, MgSO_4_ slightly reduced cerebellar hemorrhage [1]. In an original model of preterm hemorrhage, we observed a 20% reduction of hemorrhage occurrence after MgSO_4_ 600 mg/kg administration in P5 mice (not shown), in line with therapeutic trends [1,56]. As hemorrhages largely depend on brain vessel immaturity in humans, it appears likely that the deleterious vascular effects of MgSO_4_ reported for high dosages did not occur at neuroprotection low doses [8,57].

Neuroprotection in fetuses using low MgSO_4_ doses given to women at risk of preterm delivery before completed 33 GWs is now recognized and approved as the gold standard in many countries [4,58]. Its action reported in preterm exposed to MgSO_4_ in the last in utero periods showed respiratory and hemodynamic stabilization, but MgSO_4_ also showed direct neuroprotection of the neural tissue. Combined transcriptomics and metabolomics approaches showed a type of MgSO_4_ preconditioning effect on mitochondria, resulting in the sparing of high-energy phosphate and the reduction of succinate, involved in ROS production and inflammation [33].

Moving to MgSO_4_′s prevention of HI effects in the neonate mouse, a remarkable observation was that the combination of MgSO_4_ plus HI did not elicit a summation of the effects observed in separate conditions. MgSO_4_ alone had its maximum transcription effects after 6 h although it affected HI responses as early as 3 h after insult, reorientating the responses in an original direction. This effect of Mg^2+^ was likely the result of non-transcriptional effects; i.e., at protein activity levels, before Mg^2+^ induced transcription modulation occurred (Figure 6). Although HI responses largely differed in terms of identity of the genes affected in Ctrl or in MgSO_4_-treated animals, they converged toward the same functions; inflammation, innate immunity and mRNA transcription by RNA-polymerase II complex. The reduced amplitude of effects was, however, in line with expected anti-inflammatory MgSO_4_ effects [25]. Anti-inflammatory effects of MgSO_4_ at the transcription level in an HI context toward *Ccl2 Cxcl1, Ccl3*, and *Csf1* coincided with the reported effects at the protein level on early inflammation response factors (MCP-1,GRO/KC,MIP-1α and M-CSF) described in rat [33]. Transcriptomics allowed us to extend the MgSO_4_ action field to Tlr3,4,9, Il1α,β, Tnf, NfκB, and Ifng-mediated pathways. Relatedly, the absence of transcription effects on *Il4* and *Il10* transcription was in accordance with the observations at the protein level. Ifn*g* appeared as a pUR in HI at the highest significance but did not appear in MgHI, together with many pro-inflammation genes (*Stat3*, *Tlr4, Ccr2, Cx3cl1*, *Chuk, Nf**кb*, *Ccl2, Stat1,* and *Saa1*). Among these genes, only Ccl2 and Stat1 were affected at the transcription level, also indicating that MgSO_4_ essentially prevented HI-induced inflammation at post-transcription levels. *Ifng* also had a putative impact on innate immune cells since most putative downstream effectors were related to chemokine and, macrophages-immune cell communication pathways (Figure 9A).

In preterm subjects, increased Il-4 levels in the cerebrospinal fluid contributed to a cytokinic neuro-inflammatory profile [59]. In HI mice, Il-4 appeared as a pUR, suggesting an early regulatory function on the inflammation process. In Mg groups it showed a tendency to aggregate Il-4 putative downstream targets, but no clear inhibition by MgSO_4_ of HI effects indicate that Il-4 was a target of Mg. Il-4′s anti-inflammation function is in part due to its favoring the M2a macrophage/microglial phenotype in neonates [60,61]. In fact, an Mg induction of M2a phenotype has been reported as enhancing *Il4*, *Il10, Bmp2,* and *Vegf* expression [62]. In our hands, MgSO_4_ likely had post-transcription interaction with Il-4, but examination of Il-4 downstream microglial markers of repair (M2A) or the immuno-modulatory (M2B) phenotypes did not indicate convergent Mg effects [61]. Therefore, Mg^2+^’s putative anti-inflammatory effects were rather due to the inhibition of the pro-inflammatory pathways than by inducing an anti-inflammatory differentiation of microglia.

De novo effects were detected in MgHI, with high *p*-values for mitochondria and to a lesser extent for ribosome-related indexes. Magnesium reduced the transcription of genes involved in energy metabolism, protein translation, and protein degradation through ubiquitination. mTORC2 (coded by *Rictor*) and Sirt1 should be considered inhibitory hubs for oxidative phosphorylation and protein translation and degradation in MgHI mice. *Rictor* codes mTORC2, the rapamycine insensitive companion of mTOR which integrates many cell-survival-linked functions and requires Mg^2+^ for its activation [63]. MgSO_4_ previous to the Rice and Vannucci procedure in rats at P7 showed preserved mitochondrial functions after HI suggesting a pre-conditioning mechanism [33]. In our hands, putative mTORC2 MgSO_4_ activation may have resulted in the repression of 33 genes and 1 induction, of which 11 genes were coding factors of mitochondrion membrane respiration chain complex 1. These genes were distinct from those identified in rats at P7, although at a different time point, but the observations converge toward mitochondrial targeting by MgSO_4_ [33]. In addition, mTORC2 appeared to be a potential downstream hub by which Mg^2+^ affects protein translation. The repression of 12 genes coding ribosomal proteins putatively occurred downstream of mTORC2 activation, half of which were coding proteins further localized at the synapse (Rpsa, Rps1, Rps15, Rps18, Rpl7, Rpl10). Of note, none of the genes coding these proteins was affected in Mg groups in which synaptic tropism of Mg effects was noted, once again indicating a de novo mechanism recruited in MgHI.

Many MgSO_4_ pretreatment effects could be detected or suspected in the absence of up-coming HI, such as inhibitory effects on signaling pathways dependent on GPCR, a well-known target of Mg^2+^ in terms of membrane trafficking, and transduction effects. The rapid diffusion of the divalent cation and interaction at these sites support the view of rapid MgSO_4_ interference with HI, and sustain the findings of upstream regulators as putative Mg targets, even when unaffected at their own transcription level.

## 4. Materials and Methods

### 4.1. Animals

National marine research Institute (NMRI) mice, purchased from Janvier Labs (Le Genest-Saint-Isle France), were housed sex-separated at a controlled temperature (21 °C) with access to food and water ad libitum with a 12-h light/dark cycle. Reproduction was performed locally by putting females in male cages for one night to cause birth at definite days. The day of birth was noted as day 1. All nursing and experimental procedures were performed according to the recommendations of the European Communities council directives (2010/63/UE) and French national legislation. Protocols received the agreement of the French Ministry of Higher Education and Research (#01680.02/2014) and were performed by authorized experimenters with efforts to minimize animal numbers and suffering. Animal numbers are given in Appendix A.

Transcriptome studies were carried out in four groups of NMRI mice aged 5 days (P5).

The Control group (Ctrl) underwent no intervention.

The Mg group of animals received an intra-peritoneal MgSO_4_ bolus injection (3.33 µL/g) at 600 mg/kg of body weight as previously reported [28]. Brains were collected at 5 time points—1.5 h, 3 h, 6 h, 12 h, and 24 h.

The HI group in which animals were exposed to hypoxia–ischemia as previously described [28]. Briefly, isoflurane anesthetized pups had a right carotid ligature, and were returned to the mother. Then, after 60–120 min of recovery from anesthesia and surgery, the pups were exposed to 8% O_2_ for 40 min in a humidified and thermostatic chamber at 36 °C [28,37]. The end of hypoxia was considered t_0_ for the 3-h and 12-h post HI periods.

The MgHI group of animals received MgSO_4_ pretreatment one hour before being exposed to hypoxia–ischemia [28].

Behavioral studies were performed in a total of 204 mice treated at P5 by a single injection of saline buffer (PBS), MgSO_4_ (600 mg/kg), or positive control pharmacological agents—MK-801 (1 mg/kg) or ketamine (40 mg/kg), known to induce long-term effects in the behavior paradigm evaluated (details in Appendix A). Repeated daily injections of PBS or MgSO_4_ (600 mg/kg) from day 5 to day 9 were performed in separate groups. Each parameter is considered as mean ± standard error to the mean.

### 4.2. RNA Sample Preparation and Microarray Hybridization

Brains were quickly removed under isoflurane anesthesia, sectioned on medial line and ipsilateral hemispheres and were immediately frozen in liquid nitrogen. Pool samples for the microarray study were made with 5 µg of RNA extracted from 6 ipsilateral hemispheres of sex-matched injured or control mice in the different experimental groups and time points. Mice pups at each time point were taken from at least two different litters. A total of 288 pups were used for the transcriptome study (details in Appendix A).

RNA extraction was performed in individual hemispheres. Brain tissues were thawed in 350 µL of tissue lyser^®^ (Qiagen, Courtaboeuf, France) and homogenized using ceramic beads (1.4 mm Ozyme, Montigny le Bretonneaux) for 20 s at 50 Hz. Total RNA extraction was done using the Nucleospin RNA plus kit^®^ according to the manufacturer’s recommendations (Macherey-Nagel, Hoerdt, France, www.servilab.fr (accessed on 15 April 2021)) and stored at −80 °C until use. The quality and quantity of isolated RNAs were assessed using the 2100 Bioanalyzer (Agilent Technologies, Santa Clara, CA, USA) and the Nanodrop (Thermo Scientific, Wilmington, NC, USA). Labeled cRNA was synthesized from 100 ng total RNA using the Quick Amp Labeling Kit according to the manufacturer’s recommendations (Agilent Technologies, Les Ulis, France). A total of 825 ng of labeled cRNA (Cy3 for control samples and Cy5 for test samples) was co-hybridized for 17 h at 65 °C on Whole Mouse Genome Oligo 4 × 44 K microarrays (G4845A, Agilent Technologies, Les Ulis, France). Raw hybridization data were extracted using an Agilent DNA microarray scanner G2565CA (Agilent Technologies), and normalized (using LOWESS method) by Feature extraction software (v.10.5.1.1), then were transferred to Genespring^®^ (GX v.14.9.1 software, Agilent Technologies, Santa Clara, CA, USA) for data processing and data mining.

Hybridization data were filtered on spot quality (uniformity) detection level (above 2-fold background in at least 2 samples out of three in one-color analyses) and statistically selected using GeneSpring^®^ (GX 12.6 software, Agilent Technologies, Les Ulis, France), as previously described in detail [29,64]. Only probes referring to validated genes (Official Gene Symbol) were taken into account. Probes for cRIKEN sequences were excluded from analysis. In the case of multiple probes for one gene, the probe exhibiting the highest fold change was used for subsequent analyses.

Raw data were deposited in the NCBI Gene-Expression-Omnibus repository (GSE 144455: https://www.ncbi.nlm.nih.gov/geo/query/acc.cgi?acc=GSE144455 (accessed on 24 January 2020))

### 4.3. Strategy of Transcriptomic Analysis

The study was designed to extract the effects of MgSO_4_ alone (List A) and MgSO_4_ interference with HI effects (List B) at the transcriptional level. Transcription responses were measured using a two-color strategy in three independent experiments. The signal from any probe was recorded when the *p*-value in the t-test against 0 was below 5 × 10^−2^. The extraction of regulated probes was performed at each time point separately (Figure 1).
Experiment 1 (Mg vs. Ctrl): The effects of MgSO_4_ (600 mg/kg bolus at P5) were evaluated through comparison with control brains (Ctrl). The effects (inductions and repressions) were investigated at 5 time points (1 h 30 min, 3 h, 6 h, 12 h, and 24 h), in three independent samples per time point post-injection.Experiment 2 (MgHI vs. HI): The effect of MgSO_4_ pretreatment on the transcription response to HI was evaluated by comparison with HI-exposed brains at 2 time points after HI. The choice of 3 h and 12 h post-insult time points was based on previous data, in order to detect early onset and peak responses in P5 brains [29].Experiment 3 (HI vs. Ctrl): The effect of HI was evaluated by comparison to control brains at 3 h and 12 h time points to ensure reproducibility with the previous study and to provide reference effects of HI at P5 [29].

In all three experiments, the two-color analysis identified significant inductions and repressions. Thus, the effects on genes affected both in ① and ②, or in ② and ③ must take into account the direction of variation that might differ among common entities (Figure 2).

Comparisons of expression levels on one fluorochrome can also be extracted from separated arrays after normalization using the Genespring^®^ one-color protocol and allow broader comparisons of conditions. One-color analyses were performed to compare expression levels in the Mg group to (i) respective levels in the MgHI group, in order to exclude MgSO_4_ pretreatment effects that did not interfere with HI, and (ii) MgHI to Ctrl, as detailed below.
List A, extracted from the two-color analysis of experiment 1, described the transcription effects of MgSO_4_ single injection. The effects of MgSO_4_ were evaluated at 5 time points. All entities exhibiting either mean change ≥2 fold and/or *p*-values ≤ 0.05 *t* test against 0, according to Benjamini-Hochberg correction were selected at each time point in both directions and recorded in List A. Few genes showed induction and repression of separate probes. This was due to biphasic regulation (1 item) or detection of variations on separate probes (8 items). These ambiguous items were placed in induced or repressed gene lists for further analyses.
oList A_1_ grouped the fraction of List A of MgSO_4_ effects observed in the same direction in the two-color protocol in both Ctrl and HI mice, indicating HI-evoked transcription responses that were unaffected by MgSO_4_ (④ in Figure 2). The effects observed at least at one time point, in any sense of variation, at 3 and 12 h were merged in order to be handled as Mg or MgHI responses.List B reports the effects of MgSO_4_ pretreatment on transcription responses to HI. It was obtained by concatenation of three sub-lists (B_1_, B_2_, and B_3_) describing three classes of effects of MgSO_4_: (i) the effects of HI revealed after MgSO_4_ exposure (List B_1_), (ii) the reversion by MgSO_4_ of HI transcription effects (List B_2_), and (iii) the amplification by MgSO_4_ pretreatment of HI transcription effects (List B_3_). The elaboration of these lists required crossed comparisons of two-color and one-color analyses corrections and the subtraction of genes in List A_1_ (Figure 2).
oList B_1_ (concatenation of ⑤ + ⑥ + ⑦−④−⑪) contains the genes affected after HI-exposed MgSO_4_-pretreated animals in ② but not after HI in Ctrl animals in two-color analyses in ③ (⑤ in Figure 2), plus genes exhibiting opposite effects in ① and ② (⑥ in Figure 2). B_1_ was supplemented by genes exhibiting significant *t* test differences in one-color analysis between Ctrl values (in ① and ③) and MgHI values in ② (⑦ in Figure 2). The later series could reveal genes unaffected by HI (in ③). Finally, unchanged MgSO_4_ and HI effects in combined conditions (A_1_ and ⑪) were subtracted.oList B_2_ (concatenation of ⑧ + ⑨−④) contains the genes affected in opposite senses by HI in Ctrl or in MgSO_4_ pretreated animals in two-color analysis (⑧ in Figure 2), plus the genes exhibiting no difference in one-color analysis between Ctrl and MgHI (⑨ in Figure 2), minus A_1_ (④).oList B_3_ (concatenation of ⑩−④) contains the genes affected by HI in the same sense after HI in Ctrl or in MgSO_4_ pretreated animals (⑩ in Figure 2) minus A_1_ (④).List C contained genes exhibiting transcription regulation in the same sense in MgSO_4_-treated or non-treated mice in ③ and ⑦ (⑪ in Figure 2) and represented genes affected by HI but not affected by MgSO_4_ pretreatment.

### 4.4. Data Mining

Gene lists A and B, generated as detailed in Figure 2 were submitted to DAVID^®^ (v 6.8) software [65,66]. Enrichment of seKws, seGO-terms, on biological processes (GOTERM_BP_ALL; BP), cell components (GOTERM_CC_ALL; CC) and molecular functions (GOTERM_MF_ALL; MF), and seKpaths were researched using DAVID^®^ software. Keywords and GO-terms enrichment was extracted on the 1411 genes of List A, and the 1958 genes of List B filtered on gene number (N) ≥10, at the Bonferroni *p*-value threshold <1 × 10^−4^, or coincidence of *p*-value according to EASE < 1 × 10^−4^ with FE >2 [67]. Clustering of seGO-terms was performed using the online REVIGO software (http://revigo.irb.hr/ (accessed on 15 April 2021)) to discard redundancy (dispensability <0.7) [68]. Filtering conditions to extract seKPaths, were set at N ≥ 10 entities, Bonferroni *p*-value threshold < 5 × 10^−2^ and FDR < 10% for each item.

Pathway investigation was completed by the literature-based Ingenuity Pathway Analysis (IPA^®^). IPA^®^ used human gene nomenclatures (showing slight differences with mice). IPA analysis provided a z-score, indicating the sense of regulation (activation or inhibition) of significantly enriched pathways (seIPaths). We recorded the canonical pathways with N ≥ 10 and right-tailed Fisher’s Exact Test *p*-values < 0.01 and |z-score| >2, at least at a one-time point after HI or MgSO_4_ treatment. Pathway list were compared and classified using hierarchical clustering based on *p*-value and z-scores (selection set at *p*-values < 1 × 10^−4^ and |z-score| > 2), using IPA^®^, on the basis of comparison of MgSO_4_ effects at 4 different time points; HI effects at 3 h and 12 h and MgHI vs. HI effects at 3 h and 12 h, in lists determined by the Genespring^®^ normalized Cy ratio. IPA analysis did not indicate an FDR.

Putative upstream regulators (pUR) identification was performed by IPA^®^ comparisons selected based on right-tailed Fisher’s Exact Test *p*-values *p* value < 1 × 10^−4^ and |z-score| ≥ 2. List handling and presentation was undertaken using Microsoft Excel^®^ (2016 MSO 16.0.4849.1000). Comparisons of induction/repression distributions in enriched terms were performed using Fisher’ exact test using GraphPad Prism6^®^ software (La Jolla, CA, USA).

Interference of MgSO_4_ injection with development in control and in HI-exposed mice has been explored using previously described spontaneous variations of expressions before and after P5 (details in Appendix A) [37], deposited in the NCBI Gene Expression Omnibus; GSE144456).

A complement experiment of two-color analysis of MgHI and Ctrl on 3 replicates at 3 h and one at 12 h after HI was performed to consolidate one color analysis (Data in Appendix A).

### 4.5. Behavioral Studies

Three different behavioral tests were performed to assess long-term putative effects at adulthood of neonatal treatments. Six experimental groups were studied. Four groups received a single injection at P5 of saline (PBS), MgSO_4_, (600 mg/kg), MK-801(1 mg/kg), or ketamine (40 mg/kg), and two groups were treated daily for P5 to P9 with PBS or MgSO^4^. Behavior was tested at adulthood to evaluate spontaneous activity, acute MK-801 (375 µg/kg) evoked activity, and social behavior. Spontaneous activity was assessed in a free-access running wheel in the home cage for 3 days and in an open field for 30 min. In the open field, activity changes induced by acute MK-801 subcutaneous injection were recorded for 150 min separately [69]. Social interaction with an unknown congener was assessed for 6 min in a two-compartment device after a 3-min habituation period of the test animal, as previously described [37].

## 5. Conclusions

MgSO_4_ had a wide spectrum of activities slowing-down transcription, which were largely oriented toward synaptic development in P5 neonatal mice. These effects were transient and did not affect further behavioral activities in grownups. The transcription pattern response to MgHI appeared very different to the arithmetic summation of separate effects of MgSO_4_ and HI. Rather, it appeared that MgSO_4_ priming reoriented HI response in an original direction, from which neuroprotection may result. Administered before HI, MgSO_4_ reduced inflammation and innate immune responses. MgSO_4_’s slowing effects on transcription by RNA-polymerase II or on protein translation by Eif2,4 may result from the orientation of energy metabolism toward low-energy-cost high phosphate pathways. These effects possibly result from mitochondrial targeting and the reduction of protein translation but not from microglial induction toward regulatory functions.

## Figures and Tables

**Figure 1 ijms-22-04253-f001:**
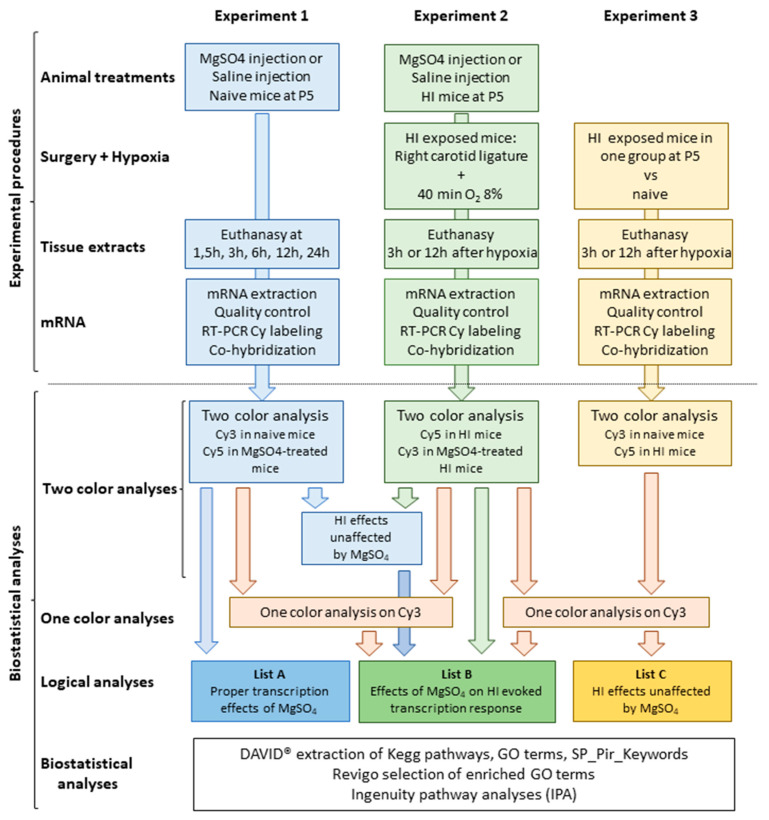
Experimental schedule. GO, Gene Ontology; HI, hypoxia–ischemia; P5, five days post-birth.

**Figure 2 ijms-22-04253-f002:**
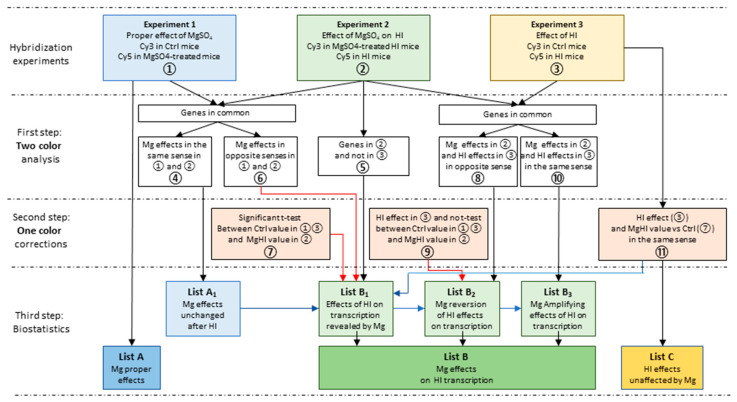
Detailed procedure of extraction of **List A** of proper effects of MgSO_4_ treatment, and **List B** collecting the different MgSO_4_ modulation of HI response to HI in P5 brain. All three experiments were analyzed independently (first step analyses in two-color co-hybridization experiments), providing lists of genes noted ①, ②, and ③. The genes detected in experiment 1 provided the List A of proper effects of MgSO_4_ single injection, to be submitted to pathway and gene ontology analyses. Coincident observations in ① and ② were extracted and split into two sub-series (④ and ⑥). **List B** constitutes the effects of MgSO_4_ seen only in the HI context (List B_1_), MgSO_4_ reversion of HI effects (List B_2_), and MgSO_4_ amplification of HI effects (List B_3_), combining two-color and one-color analyses. The subseries of List A (④)—containing genes showing induction after MgSO_4_ injection in both control (①) and HI-exposed (②) animals regrouping the genes affected by MgSO_4_ that were unchanged after HI—was subtracted from List B. **List C** contains genes affected by HI in control as well as in MgSO_4_-pretreated mice. Black and Red arrows indicate two-color and one color insertions; blue arrows indicate subtractions in destination boxes.

**Figure 3 ijms-22-04253-f003:**
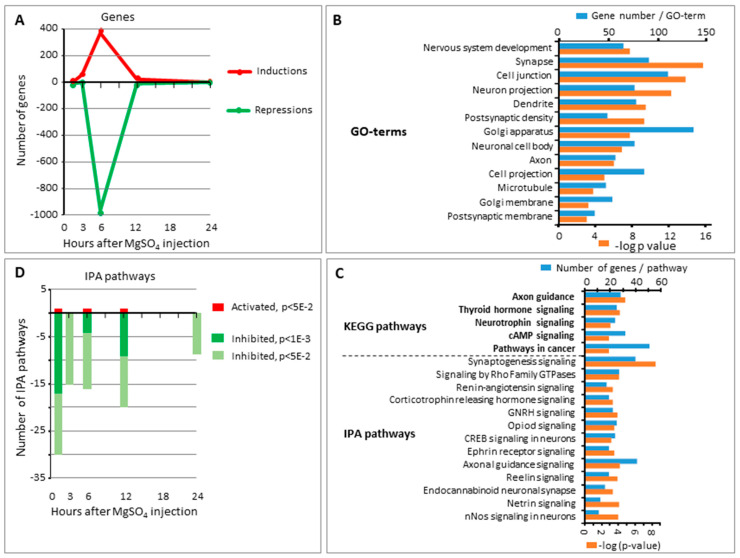
Effects of MgSO_4_ (600 mg/kg) single injection in P5 mouse brain transcription. (**A**) Time course response on inductions (in red) and repressions (in green, labeled as negative values) showing a very transient wave of inductions and repressions centered on 6 h after injection time point. (**B**) Time course distribution of significantly enriched IPA-Pathways extracted from MgSO_4_ single injection. (**C**) Gene numbers and log *p*-values in 13 Gene Ontology terms enriched from 1 h 30 min–24 h effects of MgSO_4_ single injection. (**D**) Gene numbers and log *p*-values in Kyoto Encyclopedia of Genes and Genomes (Kegg) and IPA Pathways significantly enriched from 1 h 30 min–24 h effects of MgSO_4_ single.

**Figure 4 ijms-22-04253-f004:**
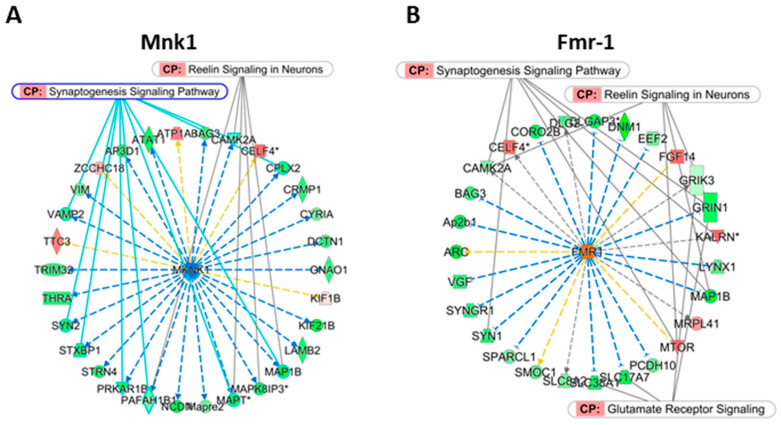
Schematic representation of two MgSO_4_ putative upstream regulators (pURs) 6 h after injection. Blue and orange background in pURs (centered) indicate inhibition and activation respectively. Red and green backgrounds in genes downstream to pURs indicate gene induction or repression. (**A**) Putative Mnk1 inhibition by MgSO_4_ based on 30 transcription effects recorded. (**B**) Putative Fmr-1 activation by MgSO_4_ based on 20 transcription effects recorded. Blue, yellow, and gray arrows indicate coherence, absence of coherence or undetermined sense of variation of respective genes in observations and in the literature, according to IPA^®^ analysis. *; multiple identifiers (probes) in data list.

**Figure 5 ijms-22-04253-f005:**
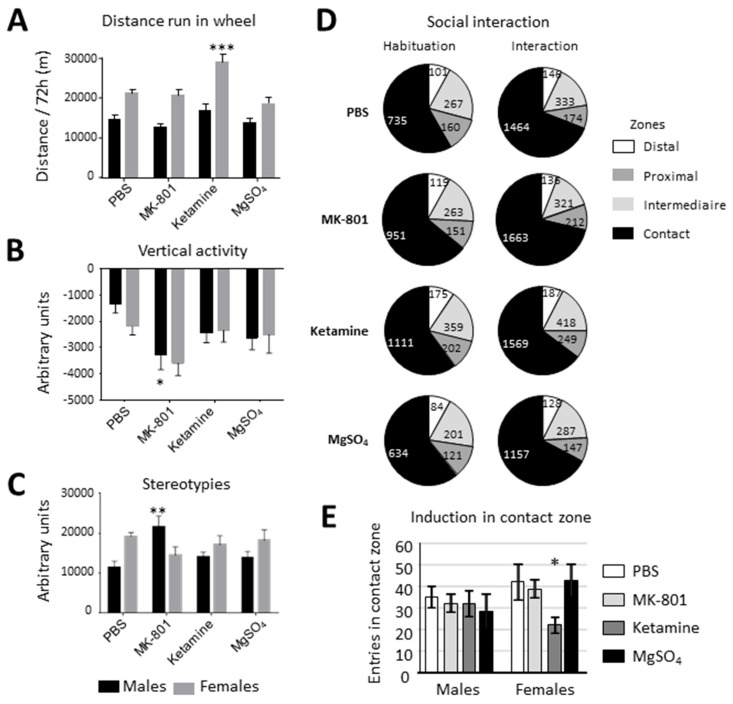
Behavioral effects in adulthood of P5 mice treated with PBS, MK-801, ketamine or MgSO_4_ at P5 in the running wheel (**A**), open field exploration (**B**), acute MK-801-evoked hyperactivity (**C**), and social interaction behavior (**D**,**E**). * *p* < 0.05, ** *p* < 0.01, *** *p* < 0.001 according to two-way ANOVA Bonferroni post-test.

**Figure 6 ijms-22-04253-f006:**
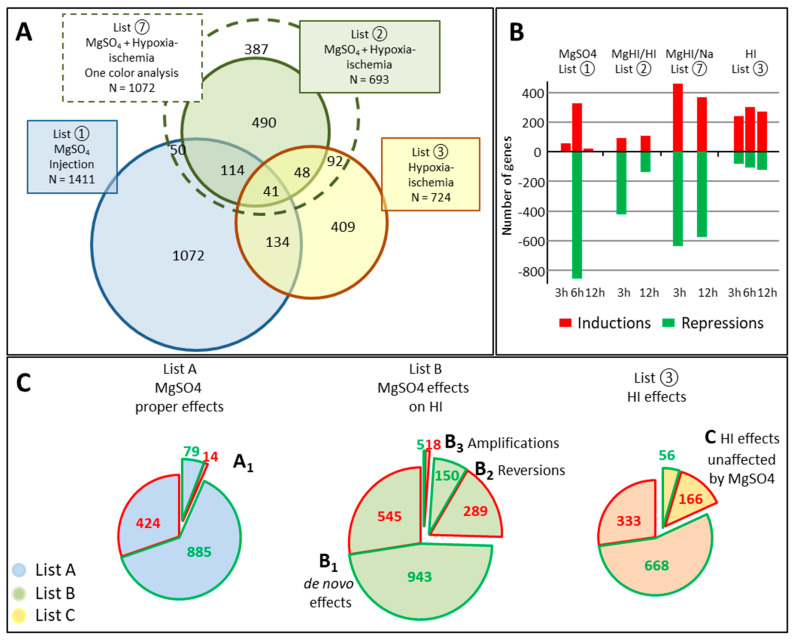
Comparative distribution of transcription effects of HI and/or MgSO_4_ in time and senses of effects. (**A**) Venn representation of coincidences in lists ①, ②, and ③. The superimposed dashed circle represents the number of genes in the one-color analysis of MgSO_4_ effects in HI (List ⑦). Note that the intersection of ①, ⑦, and ③ included only 31 genes (not shown). (**B**) Kinetics of inductions (red histograms) and repression (green histograms) at 3 h, 6 h, and 12 h after MgSO_4_, HI, or both (Lists ①, ③, and ②, respectively) from two-color analyses and MgHI effects in single-color analysis (List ⑦). (**C**) Comparison of HI effects in Ctrl ③ and in MgSO_4_-pretreated (List B) mice at P5. Color circling and numbering in red and green signify inductions and repressions, respectively. Surfaces are proportional to numbers (superposition surfaces approximate numbers). Red and green circling in List B2 of HI regulations reversed by MgSO_4_ are indicative of MgSO4 effects. Biphasic or putatively spliced genes were not taken into account in B_1_ and B_2_ representations. List numberings refer to Figure 2.

**Figure 7 ijms-22-04253-f007:**
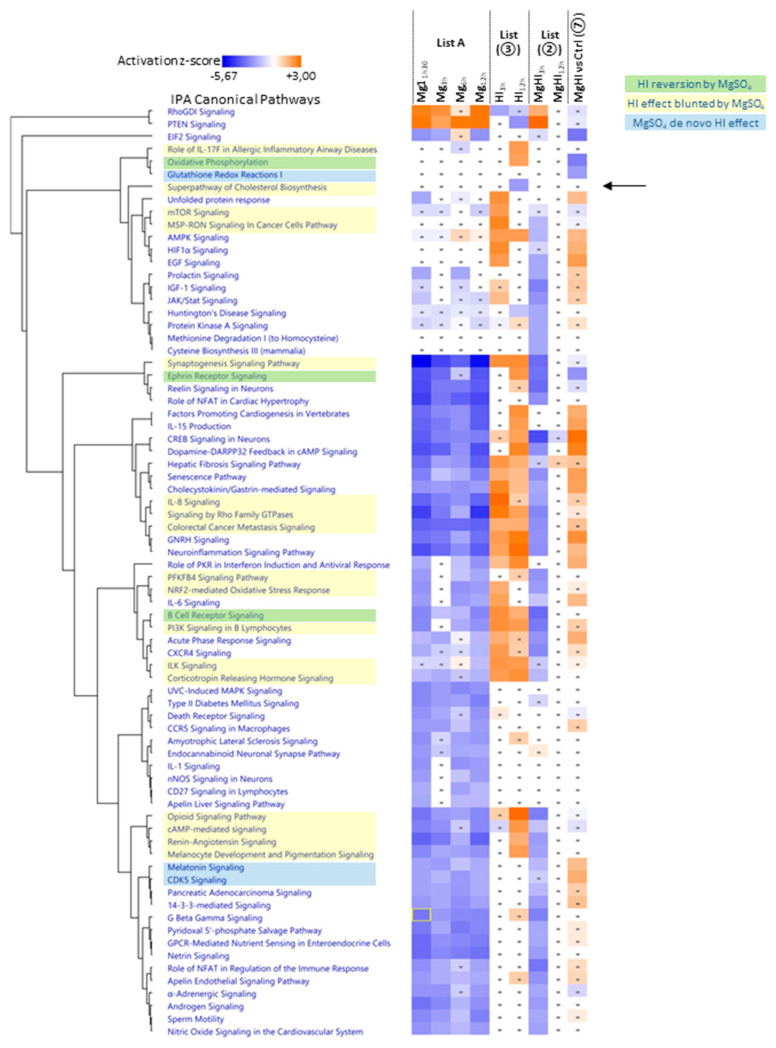
Comparative IPA analysis and time courses of pathways enrichment after HI, MgHI and in both conditions. Significant enrichment of IPA pathways was examined 3 h and 12 h after insult. Thresholds were fixed a *p*-value < 5 × 10^−2^, and z-score > |1.75|. Colored backgrounds indicate special interest seIPaths. Arrow indicates the very specific HI effect at 12 h on the superpathway of cholesterol biosynthesis.

**Figure 8 ijms-22-04253-f008:**
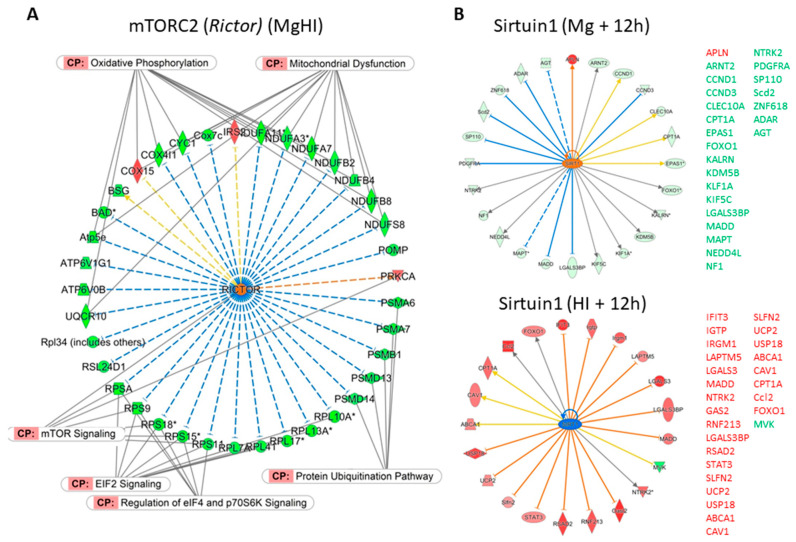
Putative upstream regulators in MgHI. (**A**) Activation of mTORC2 (coded by *Rictor*) as a pUR (*p* = 4.7 × 10^−6^, z = 4.885) as an effect of MgSO_4_ observed after HI in List B, and based on 34 gene repressions and 3 inductions. This series of gene contributed to several canonical pathways (CP) associated with *Mtor* signaling, mitochondrial dysfunction, oxidative phosphorylation, ubiquitination pathway, and protein translation by elongation factors *Eif2* and *Eif4*. (**B**) Activation of sirtuin-1 (coded by *Sirt1*) at 12 h after MgSO_4_ injection (upper panel) or inhibition 12 h after HI (lower panel) and resulting in the opposite gene repressions (in green on lists on the right) or inductions (in red on lists on the right). *; multiple identifiers (probes) in data list.

**Figure 9 ijms-22-04253-f009:**
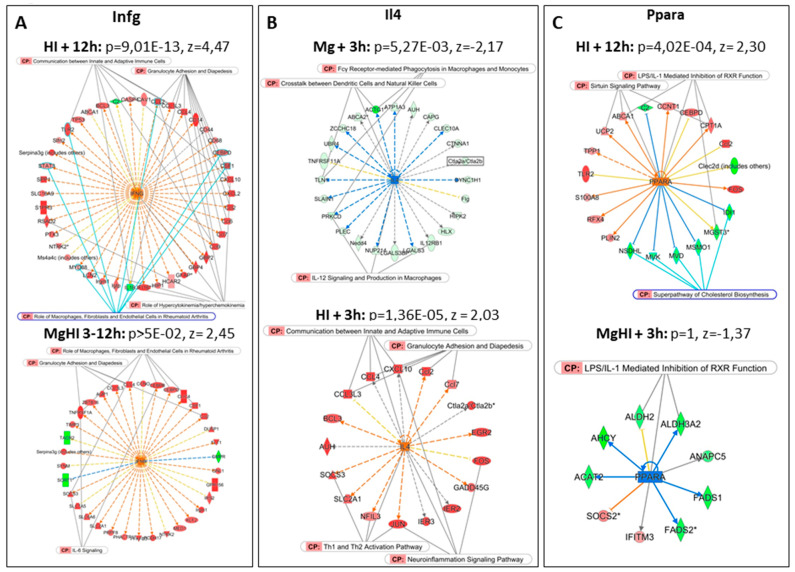
Putative upstream regulators (pURs) of MgSO_4_ effects on the HI transcription response. (**A**) Activation of Ifng appeared to be a significant activated pUR (*p* = 9.01 × 10^−13^, z = 4.472) 12 h after HI, putatively activating 42 genes also associated with inflammation and immunity CPs (top panel). In MgHI, it did not remain a significant pUR (*p* > 1 × 10^−2^, z = 3.111) although it was associated with the activation of 31 genes (bottom panel). Only 7 genes were common to the two conditions. (**B**) The inhibition of IL4 by MgSO_4_ in naïve mice identified it as a pUR (*p* = z = −3.561) associated with the repression of 24 genes, of which several were also recorded as part of CPs (top panel). *IL4* activation in the HI response makes it a pUR of inflammation activating 17 genes involved in this function (lower panel). IPA^®^ did not consider *IL4* a pUR in MgHI groups. (**C**) Activation of Ppara (*p* = 4.02 × 10^−4^, z = 2.302) in HI 12 h after insult showed it to be a pUR in the HI response in Ctrl mice, as well as the possible upstream negative regulator of cholesterol biosynthesis repressor (highlighted in blue, top panel). MgSO_4_ pretreatment reversed activation of *Ppara* with pUR inhibitory activity on function, at least at a short time post insult (bottom panel). *; multiple identifiers (probes) in data list.

**Figure 10 ijms-22-04253-f010:**
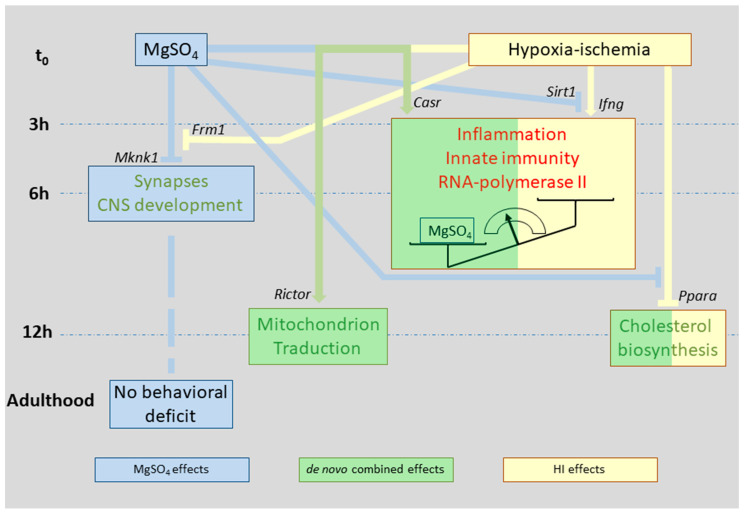
Simplified schematic effects of MgSO_4_ (in blue) and HI (in yellow) applied separately or in association (in green) with transcription in 5-day-old mouse brain hemispheres. Genes identified as putative upstream regulators of main effects are indicated. Red or green lettering indicate function activations and inhibitions, respectively.

**Table 1 ijms-22-04253-t001:** Keywords, GO terms and pathway enrichments after MgSO_4_ single injection (List A).

**Up_Keywords (David^®^ v6.8 Analysis)**	**Count**	***p*-Value ***	**Enrichment**	**FDR**	
Synapse	70	5.08 × 10^−16^	3.31	1.06 × 10^−16^	
Neurogenesis	48	3.56 × 10^−10^	3.28	3.73 × 10^−11^	
Cell junction	102	2.90 × 10^−16^	2.60	8.10 × 10^−17^	
Methylation	128	1.66 × 10^−15^	2.25	2.31 × 10^−16^	
Golgi apparatus	99	5.52 × 10^−11^	2.22	6.60 × 10^−12^	
Cell projection	82	5.09 × 10^−7^	2.04	3.55 × 10^−8^	
**GO Terms** (David^®^ analysis)	**Count**	***p*-Value ***	**Enrichment**	**FDR**	**Dispensability**
GO:0045202~synapse	91	2.69 × 10^−16^	2.80	1.87 × 10^−9^	0.00
GO:0043005~neuron projection	75	1.01 × 10^−12^	2.77	6.17 × 10^−16^	0.00
GO:0014069~postsynaptic density	49	8.14 × 10^−10^	3.18	2.32 × 10^−12^	0.04
GO:0007399~nervous system development	64	2.94 × 10^−8^	2.56	1.41 × 10^−8^	0.00
GO:0048167~regulation of synaptic plasticity	13	*4.33 × 10^−2^* ^#^	4.68	2.12 × 10^−12^	0.14
**KEGG_Pathways** (DAVID^®^ analysis)	**Count**	***p*-Value ***	**Enrichment**	**FDR**	
mmu04360:Axon guidance	28	1.66 × 10^−5^	3.27	1.36 × 10^−5^	
mmu04919:Thyroid hormone signaling pathway	25	8.09 × 10^−5^	3.30	3.30 × 10^−5^	
mmu04722:Neurotrophin signaling pathway	24	1.08 × 10^−3^	2.96	2.41 × 10^−4^	
mmu04024:cAMP signaling pathway	32	1.34 × 10^−3^	2.44	2.41 × 10^−4^	
mmu05200:Pathways in cancer	51	1.48 × 10^−3^	1.94	2.41 × 10^−4^	
mmu04910:Insulin signaling pathway	24	*4.19 × 10^−5^* ^#^	2.58	1.14 × 10^−3^	
**IPA Pathways** (Ingenuity analysis)	**Count**	***p*-Value**	**N Max ^$^**	**Kinetics** ^£^	**z-score**
Synaptogenesis Signaling Pathway	40	4.79 × 10^−9^	6 h	L (4)	−5.667
Reelin Signaling in Neurons	19	3.16 × 10^−5^	6 h	L (2)	−4.000
Axonal Guidance Signaling	41	7.08 × 10^−5^	3 h–12 h	L (2)	¤
Signaling by Rho Family GTPases	27	9.77 × 10^−5^	6 h	L (2)	−4.472
Netrin Signaling	12	1.00 × 10^−4^	6 h	L (3)	−3.317
nNOS Signaling in Neurons	11	1.12 × 10^−4^	3 h	L(2)	−2.236
GNRH Signaling	22	1.45 × 10^−4^	6 h	L(3)	−3.207
Ephrin Receptor Signaling	19	3.09 × 10^−4^	3 h–12 h	L (2)	−3.873
Opioid Signaling Pathway	25	3.63 × 10^−4^	6 h	E (1)	−3.130
Endocannabinoid Neuronal Synapse Pathway	16	4.68 × 10^−4^	6 h	E(1)	−2.324
Corticotropin Releasing Hormone Signaling	19	4.90 × 10^−4^	6 h	E(1)	−1.897
Renin-Angiotensin Signaling	17	5.01 × 10^−4^	6 h	L (3)	−3.464
CREB Signaling in Neurons	24	7.08 × 10^−4^	6 h	E (1)	−3.606
PTEN Signaling	15	8.128 × 10^−3^	12 h	L (0)	+2.714

DAVID^®^ identifications of enriched KEGG pathways and Up_Keywords, based on 1411 genes. Selection of 5 nervous system-related seGO-terms exhibiting among the highest enrichments. IPA^®^ determinations done on 1492 regulated probes exhibiting *p*-values < 1 × 10^−3^ at least at one time point, with the exception of PTEN signaling, the only seIPA with a positive z-score). * *t*-test *p*-value after Bonferroni correction (not for IPA analysis). ^$^ Max indicates the time point exhibiting the maximum number of genes in the pathway. ^#^ italics indicate *p*-value according to Fisher exact test. **^£^** Kinetics evaluated based on lowest IPA *p*-values (<1 × 10^−3^) were observed at 4 out of 5 time points (none at 24 h), noted as E (early) and L (lasting). Values in parentheses indicate the number of time points with *p*-values < 1 × 10^−3^). ¤ No z-score calculated.

**Table 2 ijms-22-04253-t002:** Gene ontology, pathway, and keyword analyses of MgSO_4_ in HI-exposed brains (List B).

**Up_Keywords (David^®^ v6.8 Analysis)**	**Count**	***p* Value ***	**Enrichment**	**FDR**	**Occurrences in Separate Lists**
Phosphoprotein	808	5.62 × 10^−36^	1.44	4.8907 × 10^−36^	③
Acetylation	417	2.01 × 10^−34^	1.81	8.7658 × 10^−35^	B1
Mitochondrion	176	1.88 × 10^−22^	2.26	5.4559 × 10^−23^	B1
Cytoplasm	483	5.33 × 10^−20^	1.49	1.1589 × 10^−20^	
Nucleus	470	4.63 × 10^−14^	1.41	1.0989 × 10^−14^	
Transit peptide	77	1.26 × 10^−6^	2.05	1.5622 × 10^−7^	B1
Ribonucleoprotein	54	2.42 × 10^−6^	2.38	2.4390 × 10^−7^	B1
Activator	88	2.52 × 10^−6^	1.91	2.4390 × 10^−7^	B1, B2, ③
Apoptosis	71	2.67 × 10^−5^	1.97	2.3265 × 10^−6^	③
Mitochondrion inner membrane	45	3.80 × 10^−5^	2.40	3.0111 × 10^−6^	B1
Ribosomal protein	38	1.07 × 10^−4^	2.54	7.7357 × 10^−6^	B1
Transport	198	1.70 × 10^−4^	1.41	1.1405 × 10^−5^	
Transcription regulation	187	4.06 × 10^−4^	1.41	2.5221 × 10^−5^	③
Ubl conjugation	161	5.29 × 10^−4^	1.45	3.0714 × 10^−5^	
Transcription	191	6.93 × 10^−4^	1.39	3.7742 × 10^−5^	
Electron transport	22	*3.06 × 10^−5 #^*	2.79	5.2884 × 10^−4^	B1
Translocation	18	*4.27 × 10^−5 #^*	3.13	7.0459 × 10^−4^	B1
**GO Terms** (David^®^ analysis)	**Count**	***p* value**	**Enrichment**	**FDR**	
BP					
GO:0006915~apoptotic process	84	*1.58 × 10^−5^^#^*	1.60	7.12 × 10^−12^	
CC					
GO:0005739~mitochondrion	282	2.61 × 10^−21^	1.80	2.53 × 10^−21^	B1
GO:0005737~cytoplasm	776	1.10 x 10^−16^	1.29	5.33 × 10^−17^	
GO:0005634~nucleus	703	1.75 × 10^−13^	1.28	4.42 × 10^−14^	B1
GO:0070062~extracellular exosome	345	1.36 × 10^−9^	1.42	3.29 × 10^−10^	
GO:0005829~cytosol	248	2.31 × 10^−9^	1.53	4.48 × 10^−10^	B1
GO:0030529~intracellular ribonucleoprotein complex	64	2.48 × 10^−6^	2.20	4.01 × 10^−7^	B1
GO:0016020~membrane	746	1.02 × 10^−5^	1.17	1.41 × 10^−6^	B1
GO:0005840~ribosome	43	2.62 × 10^−5^	2.52	3.18 × 10^−6^	B1
MF					
GO:0005515~protein binding	477	1.40 × 10^−6^	1.27	1.40 × 10^−6^	
GO:0044822~poly(A) RNA binding	161	5.94 × 10^−6^	1.57	2.97 × 10^−6^	B1
**KEGG_Pathways** (DAVID^®^ analysis)	**Count**	***p* value**	**Enrichment**	**FDR**	
mmu03010:Ribosome	30	*5.46 × 10^−5^^#^*	2.22	0.0120	B1
mmu05012:Parkinson’s disease	30	*9.13 × 10^−5 #^*	2.16	0.0120	B1
**IPA Pathways**	**Count**	***p* value**	**Max ^$^**	**Kinetics** ^£^	**z-score**
Reelin Signaling in Neurons	13	2.40 × 10^4^	3 h	3 h–12 h	−3.317
Role of NFAT in Regulation of the Immune Response	14	2.51 × 10^3^		3 h	−3.051
Molecular Mechanisms of Cancer	20	*1.05 × 10^2 #^*		3 h	¤
CXCR4 Signaling	11	*1.29 × 10^2 #^*		3 h	−2.646
Endocannabinoid Cancer Inhibition Pathway	10	*1.45 × 10^2 #^*		3 h	¤
Insulin Receptor Signaling	10	*1.51 × 10^2 #^*		3 h	−*1.897*
Clathrin-mediated Endocytosis Signaling	12	*1.58 × 10^2 #^*		3 h	¤
Clathrin-mediated Endocytosis Signaling	12	*1.58 × 10^2 #^*		3 h	¤
ILK Signaling	12	*1.62 × 10^2 #^*		3 h	¤
cAMP-mediated signaling	14	*2.14 × 10^2 #^*		3 h	¤
B Cell Receptor Signaling	12	*2.24 × 10^2 #^*		3 h	−3.317
Systemic Lupus Erythematosus Signaling	12	*2.95 × 10^2 #^*		3 h	¤
Ephrin Receptor Signaling	13	*3.24 × 10^2 #^*	3 h	3 h–12 h	−3.162
Synaptogenesis Signaling Pathway	18	*3.31 × 10^2 #^*	3 h	3 h–12 h	−3.3

DAVID^®^ identifications of enriched KEGG pathways and Up_Keywords was based on the 1958 genes, and IPA^®^ determinations were done on the 725 regulated probes of List ②. * *p*-values after Bonferroni correction, ^#^
*p*-values according to Fisher exact test. ^$^ Time point record of maximum number of genes in the pathway. ^£^ IPA enrichment was observed at the 3-h and/or 12-h time points. ¤ No z-score calculated.

## Data Availability

Data supporting the reported results can be found in the NCBI Gene-Expression-Omnibus repository; https://www.ncbi.nlm.nih.gov/geo/query/acc.cgi?acc=GSE144455 (accessed on 24 January 2020)

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
