# Peer review of "Effect of Neuroprotective Magnesium Sulfate Treatment on Brain Transcription Response to Hypoxia Ischemia in Neonate Mice"

_ijms, 2021, doi:10.3390/ijms22084253_

Round 1

Reviewer 1 Report

In this remarkable publication, the authors evaluated the effects of MgSO4 administration in the context of neonatal hypoxia-ischemia, on P5 mice. They deciphered the mechanisms of action involved. Their work led to the conclusion that the neuroprotection of MgSO4 was due, in part, to a decrease in inflammation and the innate immune response. Moreover, authors highligted a mitochondrial targeting and a reduction of protein translation.

This publication is of a very good standard, pleasant to read and of very high scientific quality. In order to complete the experimental design, the authors could have chosen a control group having undergone a simple exposure of the carotid artery, under anestia, instead of a group having undergone no intervention. In order to improve this article, the authors could (i) specify the temperature of the hypoxia chamber (because temperature is a key factor: hypothermia is neuroprotective, while hyperthermia is deleterious), (ii) could give more details on the number of animals used, by dividing them by group and type of results (either in the legends of the figures, or in the M&M, or in a table), (iii) the authors could specify if the results are expressed as an mean+/- s.e.m. or +/- SD. 

Author Response

Madam, Sir,

Thank you for your work and kind comments.

i) Indeed temperature is a key factor in subsequent consequences of insult. I apologize for this oversight. The information has been inserted in the main Material and Methods section of revised text (p. 23, l. 19).

ii) A table (Table S3) including the details of animal numbers per sex in each sub-part of the study was inserted in the supplement material.

iii) All data are presented as means ± sem. This precision has been inserted in

Material and Methods section of revised text (p. 23, l. 27).

Reviewer 2 Report

The manuscript focuses on a co-hybridization approach using the transcriptome in 5-day old mice treated with a single dose of MgSO4 (600mg/kg), and/or exposed to hypoxia-is-chemia. The transcription of hundreds of genes was altered in all the groups. MgSO4 mainly produced repressions culminating 6h after injection. The content of the paper is interesting, but there are some problem in experimental design should be solved before the manuscript been considered for publication.

Substantial revisions

Q1: Previous study has shown that 300~450mg/kg can cause all rats to die, and 174~206 mg/kg MgSO4 will reach the half-lethal dose LD50 (Mochizuki et al., 1998). In addition, Reference 26 of the MS suggested that 750~1500mg/kg can cause all rats to die. Please supplement the data result that 600 mg/kg will not cause the LD50 of Neonate Mice's half-lethal dose.

Q2: Please supplement the injection dose, volume, possible impact route, and ADEM (Absorption, Distribution, Metabolism and Elimination) MgSO4.

Q3: Please add the discussion to discuss the mechanism of HIF-1 transcription and inflammatory response on Hypoxia Ischemia after MgSO4 treatment in mice.

 Q4: Please add the discussion to discuss which pathway (IPA, KEGG? or others) may be involved in inhibiting intracellular calcium overload and brain cell death after MgSO4 treatment.

Please propose to discuss why only the ORG culture group has a lower polyphenols concentration than the CONV culture group during the same drying process in Table 1. The rest of the ingredients are the ORG culture group better than the CONV culture group?

Please suggest discussion Table 1 in the same drying process, why only training group than in CONV ORG culture group in polyphenols concentration. The remaining ingredients are better than CONV ORG culture group culture group?

Author Response

Madam, Sir,

Thank you for your work.

Q1: We are used to treat neonate mice at this dose, eventually repeatedly and never had significant mortality. In the reference indicated by the referee, Adult animals received  lasting intravenous administration of magnesium. Also, mice are different to rats (dogs) and adults  have different sensibility to drug of neonates.

Q2: MgSO4 was administered ip under a volume of 3.33 µL/g body weight. Precision have been inserted in revised text (p. 23, l. 13)

Q3: The question of HIF appears somehow marginal. In our previous work comparing transcription responses in the Rice - Vannucci HI model at P5 or P10, HIF signaling KEGG pathway appeared significantly enriched in older mice only (Dupré et al Ref 29). Several genes associated to the pathway in fact were modulated in P5 brain. The litteature described an interference of Mg ion with HIF induced Ephrin activities (Stevenson et al Plos-genetics 2012), but identified ephrin receptors different to those exhibiting modulation in our work. This interference is certainly worth investigating although, to our mind, the present data are too tenuous to be argumented at this point.

Q4: The point of Mg related cell death was evoked in the discussion. Indeed in a previous work (Daher et al 2018, Ref 28), we showed that MgSO4 administration in the same conditions as in our study did not elicit apoptosis. The referee likely thank to the work from Dribben et al, Neonatalogy 2009 supporting the view that high doses elicited apoptosis and propose the interpretation that blockade of NMDA receptor the trigger of this effect. First, the dose used by Dribben was higher that in our study. Second, our data lead to exclude that Mg had these effects at neuroprotection dose (see text p. 20, l. 15-43). Mg only high Mg doses elicited cell death as discussed in the text (p. 19, l. 19-25).

The two final comments of the referee were probably a mistake referring to another study than ours, since we  did not show culture results in this study.